# P53-Independent G1-Cell Cycle Arrest Increases SARS-CoV-2 RNA Replication

**DOI:** 10.3390/microorganisms12030443

**Published:** 2024-02-22

**Authors:** Clara Husser, Hyesoo Kwon, Klara Andersson, Sofia Appelberg, Nuria Montserrat, Ali Mirazimi, Vanessa M. Monteil

**Affiliations:** 1Department of Laboratory Medicine, Unit of Clinical Microbiology, Karolinska Institutet, 171 77 Stockholm, Sweden; clara.husser@hotmail.fr (C.H.); ali.mirazimi@ki.se (A.M.); 2National Veterinary Institute, 751 89 Uppsala, Sweden; hyesoo.kwon@sva.se; 3Biomedrex Genetics, 141 52 Huddinge, Sweden; klara.andersson@biomedrex.com; 4Department of Microbiology, Public Health Agency of Sweden, 171 65 Solna, Sweden; sofia.appelberg@folkhalsomyndigheten.se; 5University of Barcelona, 08028 Barcelona, Spain 08028 Barcelona, Spain; nmontserrat@ibecbarcelona.eu; 6Pluripotency for Organ Regeneration, Institute for Bioengineering of Catalonia (IBEC), The Barcelona Institute of Science and Technology (BIST), University of Barcelona, 08028 Barcelona, Spain; 7Centro de Investigación Biomédica en Red en Bioingeniería, Biomateriales y Nanomedicina, Institució Catalana de Recerca i Estudis Avançats (ICREA), 08010 Barcelona, Spain

**Keywords:** COVID-19, coronavirus, pathogenicity, replication, CDK2, cyclin E, CDC25A, treatments

## Abstract

While having already killed more than 7 million of people worldwide in 4 years, SARS-CoV-2, the etiological agent of COVID-19, is still circulating and evolving. Understanding the pathogenesis of the virus is of capital importance. It was shown that in vitro and in vivo infection with SARS-CoV-2 can lead to cell cycle arrest but the effect of the cell cycle arrest on the virus infection and the associated mechanisms are still unclear. By stopping cells in the G1 phase as well as targeting several pathways involved using inhibitors and small interfering RNAs, we were able to determine that the cell cycle arrest in the late G1 is beneficial for SARS-CoV-2 replication. This cell cycle arrest is independent of p53 but is dependent on the CDC25A-CDK2/cyclin E pathway. These data give a new understanding in SARS-CoV-2 pathogenesis and highlight some possible targets for the development of novel therapeutic approaches.

## 1. Introduction

Severe acute respiratory syndrome coronavirus 2 (SARS-CoV-2) is the etiological agent of Coronavirus disease 2019 (COVID-19). Patients develop flu-like symptoms that can evolve from mild to severe with lung damage but also injury to the kidneys, gastrointestinal tract, and cardiovascular system. More than 4 years after the beginning of the SARS-CoV-2-associated pandemic, the virus is still circulating at high levels worldwide and this high circulation rate favors the emergence of new variants. Despite the development of vaccines and some treatments, the virus is still causing mortality, highlighting the importance of better understanding its pathogenesis in order to develop novel antivirals.

The eukaryotic cell cycle is a meticulously coordinated and regulated series of events. It is divided into four stages: G1, during which cells prepare for DNA replication; S, during which DNA synthesis takes place; G2, during which cells prepare for division; and M, during which cells undergo mitosis. The transition between the different phases is carefully controlled by so-called cell cycle checkpoints [1]. A major cell cycle regulatory juncture is the G1 to S checkpoint, which ensures that the cell is ready to commit for cell division [2]. Hence, it is during the G1 phase that the cell integrates various signals to assess if the internal and external conditions are right for S-phase entry [3]. If those conditions are not met, the cell can exit the cell cycle into a stage called quiescence (G0). The major driving force underlying cell cycle progression is the periodic synthesis and degradation of cyclins and their association with cyclin-dependent kinases (CDK) [1,4]. Each cyclin/CDK complex relates to a particular phase and transition of the cell cycle. To be activated, CDKs must be appropriately phosphorylated and must be in complex with the appropriate cyclin. When active, cyclin/CDK complexes phosphorylate key proteins of the cell cycle, including the retinoblastoma protein (pRb) [5]. Hypophosphorylated pRb exerts an inhibitory effect on the cell cycle by binding and regulating the transcription factor E2F [5]. In G1, active cyclin D/CDK4,6 and cyclin E/CDK2 complexes phosphorylate pRb. The resulting hyperphosphorylated form of pRb releases E2F and allows the expression of S-phase genes and the progression of the cell cycle [2,4,5]. Cyclin/CDK complexes are regulated by the cyclin-dependent kinase inhibitors (CDKi). CDKis can be divided into two families, the INK4 family (p15, p16, p18, p19), and the CIP/Kip family (p21, p27, p57) [2,6].

Viruses rely on host cell factors to replicate their genomes and generate new progeny viruses. Because of this dependence, viruses have evolved a myriad of mechanisms for exploiting host cell functions. Many viruses subvert the host cell cycle in order to increase their replication [7,8,9]. For example, severe acute respiratory syndrome coronavirus (SARS-CoV) and the mouse hepatitis virus (MHV) are known to induce a cell cycle arrest in infected host cells [10,11,12,13]. Given that SARS-CoV and SARS-CoV-2 are from the same species, it is reasonable to hypothesize that SARS-CoV-2 can also initiate cell cycle arrest. In vitro experiments using transfection of plasmid coding for SARS-CoV-2 viral proteins showed SARS-CoV-2 proteins N can induce a G0/G1 [14] or S-phase [15]. Infection experiments showed SARS-CoV-2 can causes cell cycle arrest in S [15,16] and G2/M stages [16]. All these studies suggest that SARS-CoV-2 could take advantage of the modulation of the cell cycle [15,17].

Understanding how different cell cycle phases are beneficial for SARS-CoV-2 pathogenesis could highlight some possible targets for the development of novel therapeutic approaches [18].

In this study, we show that the cell cycle arrest in late G1 is beneficial for SARS-CoV-2 replication. This cell cycle arrest is independent of p53 but is dependent on the cell division cycle 25 A (CDC25A)-CDK2/cyclin E pathway.

## 2. Materials and Methods

### 2.1. Cells and Virus

SARS-CoV-2 (GenBank accession number MT093571) was isolated and propagated on Vero-E6 cells (ATCC CRL-1586) as previously described [19]. A549 (ATCC CCL-185) were cultured on DMEM containing 10% FBS. Serum starvation to induce quiescence or synchronization of cells was performed by culturing cells in 0% FBS medium for 24 h.

Kidney organoids were prepared as previously described [19] and cultured in RPMI (ThermoFisher Scientific, Waltham, MA, USA). 

Briefly, human embryonic stem cells (The National Bank of Stem Cells (ISCIII, Madrid, Spain)) were grown on vitronectin-coated plates (1001-015, Life Technologies, Carlsbad, CA, USA) and incubated with 0.5 mM EDTA (Merck, Darmstadt, Germany) at 37 °C for 3 min for disaggregation. In total, 100,000 cells/well were plated on a 24-well plate coated with 5 μL/mL vitronectin and further incubated with supplemented Essential 8 Basal medium at 37 °C overnight. The day after, named as day 0, the cells were treated for 3 days in Advanced RPMI 1640 basal medium (ThermoFisher Scientific) supplemented with 8 μM CHIR (Merck) and 1% penicillin-streptomycin and 1% of GlutaMAX (ThermoFisher Scientific). The medium was changed every day. From day 3 to 4, media were changed to Advanced RPMI supplemented with 200 ng/mL FGF9 (Peprotech, Cranbury, NJ, USA), 1 μg/mL heparin (Merck), and 10 ng/mL activin A (R&D System, Minneapolis, MN, USA). On day 4, the cultures were rinsed twice with PBS, and resuspended in Advanced RPMI supplemented with 5 μM CHIR, 200 ng/mL FGF9, and 1 μg/mL heparin. Cellular suspensions were seeded in a V-shape 96 multi-well plate at a final concentration of 100,000 cells/well and centrifuged at 2000 rpm for 3 min. The resulting spheroids were incubated during 1 h at 37 °C. Culture media was replaced by Advanced RPMI supplemented with 200 ng/mL FGF9 and 1 μg/mL heparin for 7 additional days. The media was changed every second day. From day 11 to 16, developing organoids were incubated only in the presence of Advanced RPMI, the media was changed every second day.

Lung organoids were kindly gifted by Prof. Haibo Zhang. The maturation media for lung organoids contained 75% IMDM, 25% Ham’s F-12, 0.5% N2 supplement, 1% B27 supplement, 0.75% of BSA 7.5%, 1% PenStrep, 1% Glutamax and 3 µM CHIR99021, 10 ng/mL FGF10, 10 ng/mL FGF7, 50 nM Dexamethasone, 0.1 mM 8-Bromo-cAMP, 0.1 mM IBMX, 50 µg/mL ascorbic acid, and 0.4 µM monothioglycerol.

### 2.2. Treatment with EIPA

A549 cells were seeded at a density of 5 × 10^4^ cells per well in a 48-well plate (Viability and infection assays) or at a density of 2 × 10^5^ cells per well in a 6-well plate (Flow cytometry and Western blot assays). At 24 h post seeding, cells were treated with EIPA or mock-treated (DMSO 0.4%) at the indicated concentration for 24 h. Cells were then analyzed for cell cycle distribution by flow cytometry, for viability using a LIVE/DEAD cell imaging kit (#R37601, ThermoFisher Scientific), for Western blot or for infection as described below.

### 2.3. Viability/Cytotoxicity Assay

Mock and EIPA-treated cells were washed twice with PBS and 100 µL of fresh PBS was added per well. Reagents were thawed and mixed. In total, 100 µL of the reagent solution was added on top of the cells. The plate was incubated for 15 min at room temperature. Images of the cells were captured using confocal laser scanning microscope (Zeiss LSM 800) at 488/515 nm for live cells (green) and 570/602 nm for dead cells (red).

### 2.4. Western-Blot

Cells were washed once with PBS before being detached using 500 µL of trypsin. Cells were incubated for 5 min at 37 °C, allowing the cells to detach. A total of 500 µL of DMEM 10%FBS was added to stop the action of trypsin and the cells were transferred to 1.5 mL tube. Cells were pelleted by centrifugation at 400× *g* for 5 min. The supernatants were discarded and total proteins were extracted by lysing the cells with an in-house lysis buffer (pH 7.5) containing 10 mM Tris–HCl, 150 mM NaCl, 0.5% SDS, and 1% Triton X-100 supplemented with complete Protease Inhibitor Cocktail Tablets (Roche, Indianapolis, IN, USA) and NuPAGE LDS Sample Buffer (ThermoFisher Scientific) and β-Mercaptoethanol (Invitrogen, Waltham, MA, USA). The samples were boiled for 20 min at 98 °C. Total protein concentration was measured using Thermo Scientific Ionic Detergent Compatibility Reagent for Pierce 660 nm Protein Assay Reagent and Thermo Scientific Pierce Bovine Serum Albumin Standard Pre-Diluted Set as standard. For Western blot analyses, 20 μg of protein was separated in Criterion™ XT 4–12% Bis-Tris gels with XT MOPS running buffer and blotted onto PVDF membrane Trans-Blot Turbo Midi 0.2 μm (BioRad, Hercules, CA, USA). The membranes were blocked at room temperature for 1 h with 5% milk in PBST (PBS 0.01M + 0.1% Tween20). The membranes were then incubated with primary antibodies diluted in 5% milk/PBST for 1 h under rocking at room temperature. The membranes were then washed with PBST three times and then incubated with the secondary antibody diluted in 5% milk/PBST. After washing three times with PBST and once with PBS 1X, membrane-bound antibodies were detected by chemiluminescence using Amersham ECL Prime Western Blotting Detection Reagent (Cytiva, Marlborough, MA, USA).

Antibodies used: p-pRb (1/1000, #702097, ThermoFisher Scientific), p53 (1/1000, #MA5-12557, ThermoFisher Scientific), β-actin (1/2000, #MA5-15452, ThermoFisher Scientific), ACE2 (1/5000, #SAB3500346, Sigma), CDC25A (1/1000, PA5-77902, SAB3500346), Calnexin (1/10000, in-house), and cyclin E (1/1000, #11554-1-AP, ThermoFisher Scientific). Horseradish Peroxidase (HRP)-conjugate secondary antibody (AffiniPure Goat anti-rabbit (1/5000; #111-035-003) and/or AffiniPure goat anti-mouse (1/10,000; #115-035-174) (Jackson ImmunoResearch, West Grove, PA, USA) and Streptactine HRP-conjugate (#1610380, Bio-Rad) for ladder staining.

### 2.5. Flow Cytometry

The treated cells were detached using trypsin (ThermoFisher Scientific) (500 µL per well of a 6-well plate) for 5 min at 37 °C. A total of 500 µL of DMEM 10% FBS was added to each well and the cells were transferred to 15 mL tubes. The cells were centrifuged at 400× *g* for 5 min to pellet them. The supernatant was carefully removed by pipetting. The cells were fixed in suspension by carefully adding chilled ethanol 70% and were vortexed before being incubated for 30 min at 4 °C. The cells were centrifuged at 400× *g* for 5 min. Ethanol was carefully discarded by pipetting. The cells were washed twice with PBS, being centrifuged at 400× *g* for 5 min after each washing and PBS carefully discarded. The cells were treated 50 mL of RNase A (100 µg/mL) and stained with 200 µL of propidium iodide (50 µg/mL, ThermoFisher Scientific) for 30 min at 4 °C in the dark. The cells were then analyzed on BD FACS CantoII (BD) and data were analyzed using ModFit LT 6.0 (Verity Software, Noida, Uttar Pradesh, India).

### 2.6. Treatments

A549 cells were seeded at a density of 5 × 10^4^ cells per well in a 48-well plate in DMEM 5% FBS. At 24 h post-seeding, the supernatants were removed. The cells were washed once with PBS and the cells were treated with the respective concentration of compound:

Palbociclib 1 µM, Abemaciclib 1 µM, Gefitinib 25 µM, Cediranib 3 µM, Resveratrol 100 µM, Hesperidin 100 µM, AUZ-454 1 µM, 10 µM, and 20 µM in DMEM 5% FBS. All compounds were purchased from MedChem Express. After 24 h, supernatants were discarded. The cells were washed once with PBS and the cells were infected with SARS-CoV-2 as described below. EIPA treatment was used as a positive control and DMSO 0.4% as a negative (mock-treatment).

### 2.7. siRNA Transfection

In total, 5 pmol of negative control siRNA (Qiagen, Venlo, The Netherlands) or 5 pmol of human cyclin E1 siRNA (Horizon Discovery, Waterbeach, UK) were mixed in Opti-MEM (ThermoFisher Scientific) with lipofectamine RNAiMax (ThermoFisher Scientific) according to the company instructions. Briefly, 50 µL of the mix siRNA/lipofectamine were put in wells of a 48-well plate and 5 × 10^4^ A549 cells were added on top (200 µL in DMEM 5%FBS). Cells were incubated at 37 °C for 48 h before being infected. The protocol was similar for the cells used for Western blot, using 25 pmol of siRNA in 250 µL of opti-MEM/lipofectamine RNAiMAX and adding 2 × 10^5^ cells on top in 2 mL of DMEM 5% FBS and incubated for 48 h.

### 2.8. Infection

Cells were washed once with PBS. PBS was removed and all samples were infected for 1 h with SARS-CoV-2 (100 µL/well) at an MOI of 0.1 in 2% FBS medium containing the different compounds. At 1 hpi, the cells were then washed once with PBS and fresh medium 5%FBS containing the compounds. At 24 hpi, supernatant was removed. The cells were washed 3 times with PBS and all cells were lyzed using 100 µL per well of trizol (ThermoFisher Scientific).

Kidney organoids were infected with 10^3^ PFU/organoids for 3 days or 10^6^ PFU for 1 day in RPMI medium containing 50 µM of EIPA. Lung organoids were infected with 10^6^ PFU for 3 days in maturation media containing EIPA under shaking. 

### 2.9. RNA Extraction and RT-PCR Analysis

RNA was isolated from infected cells using Trizol (Invitrogen). Total RNA extraction from cells was performed using the Direct-zol RNA Miniprep kit (Zymo Research, Irvine, CA, USA). Equal amounts of Trizol and ethanol 100% were added to each sample and mixed. The samples were then transferred on extraction columns and centrifuged at 12,000× *g* for 30 s. The flow-through were discarded in chemical waste for Trizol. A total of 400 µL of pre-wash buffer was added on top of each column and columns were centrifuged at 12,000× *g* for 30 s. The flow-through were discarded in chemical waste for alcohol and this step was repeated once. In total, 700 µL of washing buffer was added to each column and columns were centrifuged at 12,000× *g* for 2 min. Columns were transferred to 1.5 mL tubes and RNA was extracted by the addition of 30 µL of DNAse/RNase-free water on top of each column. The columns were centrifuged at 12,000× *g* for 30 s. The columns were discarded and the eluted RNA was stored at −80 °C until qRT-PCR.

qRT-PCR mix was performed in 20 µL final volume in LightCycler Capillaries (Roche, Indianapolis, IN, USA) containing for SARS-CoV-2, 0.8 µL of forward primer (10 µM), 0.8 µL of reverse primer (10 µM), 0.4 µL of probe (10 µM), 5 µL of TaqMan Fast Virus 1-step master Mix (ThermoFisher Scientific), 5 µL of RNA and 8 µL of DNase/RNase-free water. For RNase P, 1 µL of forward primer (10 µM), 1µL of reverse primer (10 µM), 0.4 µL of probe (10 µM), 5 µL of TaqMan Fast Virus 1-step Master Mix (ThermoFisher Scientific), 5 µL of RNA, and 7.6 µL of DNase/RNase-free water.

qRT-PCR amplification was performed in a Roche LightCycler 4.0 Instrument under the following conditions: reverse transcription, 10 min at 50 °C; denaturation, 2 min at 95 °C; amplification 45 cycles of 10 s at 95 °C for the denaturation and 40 s at 60 °C for the annealing/extension step.

SARS-CoV-2 infection levels were calculated using E-gene SARS-CoV-2 primers/probe.

Forward primer: 5′-ACAGGTACGTTAATAGTTAATAGCGT-3′Reverse primer: 5′-ATATTGCAGCAGTACGCACACA-3′Probe: FAM-ACACTAGCCATCCTTACTGCGCTTCG-MGB

RNase P was used as an endogenous gene control to normalize the levels of intracellular viral RNA.

Forward primer: 5’-AGATTTGGACCTGCGAGCG-3’Reverse primer: 5’-GAGCGGCTGTCTCCACAAGT-3’Probe: FAM-TTCTGACCTGAAGGCTCTGCGCG-MGB

### 2.10. Statistical Analysis

All statistical analysis were performed using GraphPad Prism 10.

## 3. Results

### 3.1. Cell Cycle Arrest in G0/G1 P53-Independent Improves SARS-CoV-2 Infection

5-(N-Ethyl-N-isopropyl)amiloride (EIPA), an inhibitor of macropinocytosis, was used to treat A549 cells, an adenocarcinoma-derived human alveolar basal epithelial cell line. A549 cells were treated with 50 µM of EIPA or DMEM 0.4% (mock) in DMEM 5% for 24 h. These cells were then infected with SARS-CoV-2 (MOI 0.1) in medium containing EIPA. Surprisingly, 24 h post-infection (hpi), EIPA-treated cells showed a significant increase in SARS-CoV-2 infection compared to the mock-treated cells (Figure 1A). To determine if this effect was specific to SARS-CoV-2, the cells were treated and infected with another RNA virus, Hazara virus (HAZV), under the same conditions. As shown in Figure 1B, EIPA treatment was instead deleterious for HAZV infection. These data show that treatment with EIPA made A549 cells more susceptible to SARS-CoV-2 infection and this effect was specific to SARS-CoV-2.

Interestingly, EIPA treatment (50 µM, 24 h) of A549 cells visibly reduced the number of cells (Figure 2A, upper panel). According to the lack of dead cells in the EIPA-treated wells (Figure 2A lower panel), EIPA treatment was not cytotoxic. These data indicate that the reduced cell number seemed to be linked to a stop in cell growth. EIPA was previously shown to induce a cell cycle arrest in G0/G1 in MKN28 cells [20], as well as in A549 [21]. Cell cycle distribution in mock and EIPA-treated cells was assessed by flow cytometry. As shown in Figure 2B, EIPA treatment led to an increase in the cell number in G0/G1 and G2 stage compared to the S stage, with most of the cells being stopped in the G0/G1 phase. Western blot analysis of mock and EIPA-treated cells showed a dose-dependent low phosphorylation of the retinoblastoma protein (pRb), a marker of the G0/G1 stage, under treatment with EIPA, confirming that most of the cells were stopped in the G0/G1 stage (Figure 2C). Interestingly, when looking at the level of p53, a protein involved in the regulation of the cell proliferation, p53 did not show any variation under EIPA treatment, highlighting that the observed cell cycle arrest is p53-independent (Figure 2C).

To confirm that the observed cell cycle arrest in G0/G1 is responsible for the increased SARS-CoV-2 infection, A549 cells were FBS starved for 24 h before being infected with SARS-CoV-2. Starvation, leading to a cell cycle arrest in G0/G1, led to an increase in SARS-CoV-2 infection (Figure 3), as previously observed in EIPA treated cells (Figure 1A).

### 3.2. SARS-CoV-2 Entry Step Is Not Affected by the Cell Cycle Arrest

The observed increase in SARS-CoV-2 infection in A549 cells treated with EIPA could be due to a higher infection at the attachment/entry step or higher RNA replication rate. To distinguish between these alternatives, EIPA and mock-treated cells were infected with SARS-CoV-2 (MOI 0.1) for 1h and the cells were recovered 1hpi. As shown in Figure 4A, EIPA treatment did not affect SARS-CoV-2 attachment/entry step. To complete these data, A549 cells starved/non starved and treated with several concentration of EIPA were tested for ACE2 expression, ACE2 being the main SARS-CoV-2 receptor [22,23,24] by Western blot. Figure 4B shows that A549 does not express ACE2 or at an undetectable level and this level of expression is not improved when the cells are treated with EIPA, confirming the absence of role for the entry step in the observed increased in SARS-CoV-2 infection.

### 3.3. Late G1 p53-Independent Cell Cycle Arrest Leads to an Increase in SARS-CoV-2 RNA Replication

Having found that G0/G1 cell cycle arrest was responsible for the increasing in SARS-CoV-2 RNA replication, we next focused on the cellular pathways involved in the G0/G1 stage (Figure 5A) in order to understand which pathways are important for SARS-CoV-2 replication. To do this, we used several compounds that are known to block the cell cycle in G0/G1 in A549 cells (Palbociclib [25], Abemaciclib [26,27], Gefitinib [28], Cediranib [29], Resveratrol [30], and Hesperidin [31]). These compounds target different pathways described in Figure 5B. Inhibitors of growth factor receptors (EGFR and VEGFR) tyrosine kinase (Gefitinib and Cediranib) led to an increase in SARS-CoV-2 infection (Figure 5C). Interestingly, Resveratrol, which blocks the cell cycle in a p53-independent manner, also positively affected SARS-CoV-2 infection while Hesperidin, blocking the cell cycle in a p53-dependent manner, did not affect SARS-CoV-2 infection (Figure 5C). These data are consistent with the non-variation in p53 expression in the EIPA-treated cells that was previously observed (Figure 2C), which was already suggestive of a p53-independent role of G0/G1 in SARS-CoV-2 infection. Interestingly, while Palbociclib, an inhibitor of CDK4/6, had no effect on SARS-CoV-2 infection, Abemaciclib, which is also an inhibitor of CDK4/6 but also of CDK2/cyclin E, increased SARS-CoV-2 infection (Figure 5C). These data show that the increased SARS-CoV-2 infection was not caused by blockade of the CDK4/6 pathway, i.e. early G1 stage, but was dependent on the blockade at the late G1 stage via the CDK2/cyclin E pathway.

### 3.4. Cell Cycle Arrest at the CDK2/Cyclin E Checkpoint via CDC25A Degradation Is Responsible for Higher Levels of SARS-CoV-2 Infection

Next, we wished to confirm the involvement of CDK2 in the enhancement of SARS-CoV-2 infection. Since the knock-down or knock-out of CDK2 does not affect cell proliferation [32], we instead treated cells with an inhibitor of CDK2 (AUZ 454) for 24 h before being infected with SARS-CoV-2. Treatment with the CDK2 inhibitor led to an increase in SARS-CoV-2 infection, (Figure 6A). Since CDK2 and cyclin E are in complex and modulate the G1/S checkpoint, we also assessed the role of cyclin E in SARS-CoV-2 infection. Cyclin E was knocked down using siRNA (Figure 6B). The knock-down of cyclin E led to a significant increase in SARS-CoV-2 infection (Figure 6B). These data confirm the beneficial role of the inhibition of CDK2/cyclin E for SARS-CoV-2 replication.

In order for the cells to pass from the G1 to the S phase, CDK2 has to be dephosphorylated by the cell division cycle 25A (CDC25A) phosphatase (Figure 6C). Under genotoxic insults, CDC25A is ubiquitinated by Checkpoint kinase 1 and 2 (CHK1 and CHK2) and is degraded by the proteasome. As a result, CDK2 is not dephosphorylated and cells stop in the G1 phase (Figure 6C). To determine if CDC25A is involved in the blocking of cells in the G1 stage while treated with EIPA, A549 cells were treated with different concentrations of EIPA for 24 h and the expression level of CDC25A was defined by Western blot. As shown in Figure 6D, CDC25A was degraded in a dose-dependent manner, confirming the role of the degradation of CDC25A in the blocking of the cell cycle in G1.

### 3.5. SARS-CoV-2 Infection Is Also Increased in Treated Human Kidney and Lung Organoids

Since A549 cells are a cancer-derived cell line, the validation of the findings in a more relevant model is essential. To achieve this, kidney as well as lung organoids were produced as previously described [19,33]. Kidney and lung organoids were treated with EIPA for 24 h before being infected with SARS-CoV-2 for 1 or 3 days. Consistent with our results in A549 cells, EIPA treatment led to a significant increase in SARS-CoV-2 infection in kidney (Figure 7A,B) and in lung (Figure 7C) organoids, suggesting EIPA treatment leads to a similar effect in organoids and in cancer cells.

## 4. Discussion

On 31st of December 2023, SARS-CoV-2 was officially responsible for more than 770 million COVID-19 cases and 7 million deaths (WHO dashboard, https://data.who.int/dashboard/covid19/, accessed on 31 January 2024). The virus is still circulating and evolving, making the understanding of its pathogenesis essential. 

Several studies have previously highlighted the role of p53 in SARS-CoV-2 infection [34,35], pointing out p53 as a potential target for the development of antivirals [36,37]. In contrast, our data show that the increase in SARS-CoV-2 replication observed when the cells are stopped in G1 is p53-independent, making the use of p53 targeting antivirals ineffective in this case. These data highlight the importance of finding several pathways used by SARS-CoV-2 in order to develop and use the appropriate antivirals to each situation.

Our data show that serum-starved cells and cells stopped in late G1 via CDK2/cyclin E inhibition leads to an increase in SARS-CoV-2 replication. It is possible to debate whether our experiments induce a G0 or G1 cell cycle arrest as it is commonly described that serum-starved cells are quiescent and stopped in an “out of cell cycle” stage named G0. However, G0 cells have the same amount of DNA as cells arrested in G1 and this cell cycle stage is commonly called G0/G1 as they cannot be distinguished. Some suggest that the G0 stage is not a bona fide cell cycle stage but instead a G1 stage where the cells are growing so slowly that they appear to be arrested. The aim of this study is not to debate the G0/G1 cell cycle definition, but a review discussing these terms is available [38].

We showed that blocking growth factor receptor (EGFR and VGFR) tyrosine kinases is beneficial for SARS-CoV-2 infection. Interestingly, a study using several inhibitors targeting several pathways downstream of growth factor receptor, PI3K, and MAPK pathways, showed an inhibition of SARS-CoV-2 replication [39]. None of the targets were in the cell cycle pathway we highlighted today (EGFR-(most probably Ras-Raf-MEK-ERK)-CDC25A-CDK2/cyclin E). This fact exemplifies the critical importance of studying in depth the mechanisms involved in SARS-CoV-2 (and any virus) pathogenesis to properly target the right pathway depending on the context of the infection.

Several studies have previously shown the ability of SARS-CoV-2 to drive cell cycle arrest in different phases [15,16]. We found that degradation of CDC25A, which inhibits CDK2 via the lack of its dephosphorylation, and treatment with a CDK2 inhibitor, both lead to an increase in SARS-CoV-2 replication. It was previously shown that the SARS-CoV-2 N protein is able to induce a Smad3-dependent cell cycle arrest in G1, via its interaction with Smad3, leading to acute kidney injury (AKI) [14]. Smad3 (Mothers against decapentaplegic homolog 3) protein, a transcription factor/tumor suppressor in the transforming growth factor–beta (TGF-β) signaling, triggers the cell death pathway in acute kidney injury [40] and induces a G1 cell cycle arrest in cells [41]. Smad3-/- mice exhibit metastatic colon cancer [42] and the loss of Smad3 expression increases susceptibility to tumorigenicity in human gastric cancer [43]. Importantly, Smad3 mediates degradation of CDC25A via the ubiquitination of CDC25A [44], blocking the cells in G1. All these data are consistent with our results and highlight the important role of the CDC25A/CDK2/cyclin E pathway-dependent G1 cell cycle arrest in SARS-CoV-2 infection. Further investigations will be conducted to understand how this CDC25A/CDK2/cyclin E pathway affects SARS-CoV-2 replication.

## 5. Conclusions

Studies have suggested that SARS-CoV-2 is able to stop the cell cycle of infected cells [14,16], but the effect of cell cycle arrest on SARS-CoV-2 infection is unknown. Our study showed that the late G1 cell cycle arrest dependent of the CDC25A/CDK2/cyclin E pathway and independent of P53, which is able to improve SARS-CoV-2 RNA replication. 

## Figures and Tables

**Figure 1 microorganisms-12-00443-f001:**
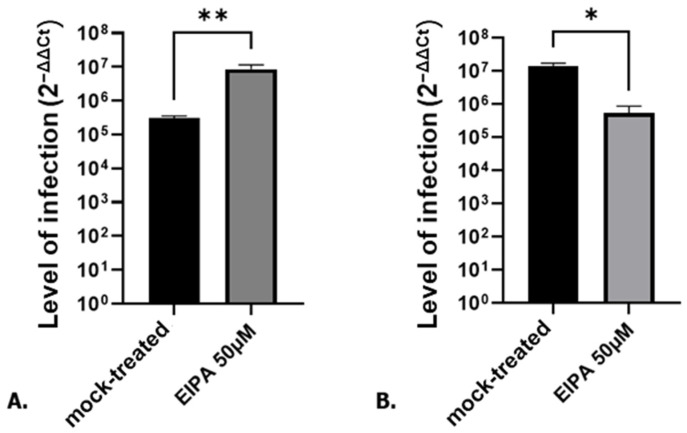
SARS-CoV-2 (**A**) and HAZV (**B**) infection level (2^−ΔΔCt^) in mock-treated or EIPA-treated A549 cells. Student *t*-test * *p* < 0.05, ** *p* < 0.01.

**Figure 2 microorganisms-12-00443-f002:**
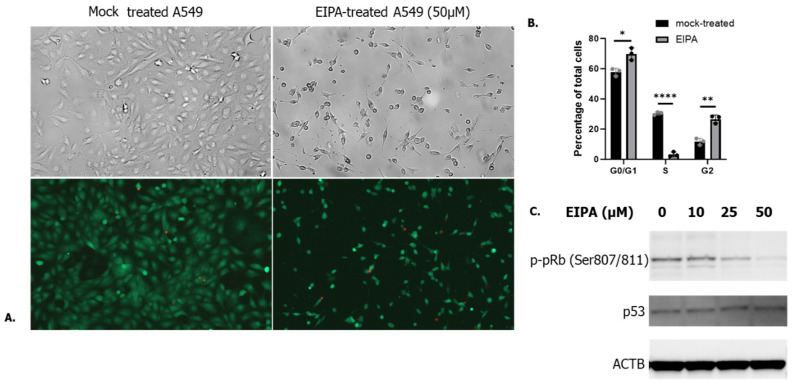
(**A**) Upper: Microscopy analysis of the mock-treated and EIPA-treated cells. Lower: A549 cells were stained for live (green) and dead (red) cells. Magnification 10×. (**B**) Cell cycle distribution of cells mock or EIPA-treated. Student *t*-test * *p* < 0.05, ** *p* < 0.01, **** *p* < 0.0001. (**C**) Staining of phosphorylated pRb and p53 by Western blot. Phosphorylation of pRb led to cell cycle progression from G1 to S. Lack of pRb phosphorylation is a marker of cell cycle arrest.

**Figure 3 microorganisms-12-00443-f003:**
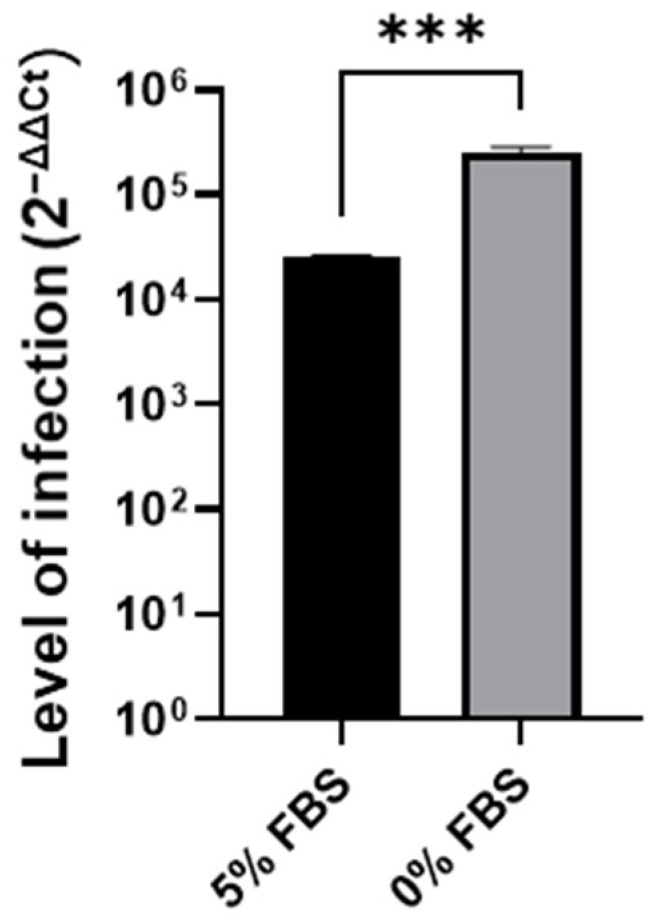
SARS-CoV-2 infection level in normal FBS (5%) or starved (0%) A549 cells. Student *t*-test *** *p* < 0.001.

**Figure 4 microorganisms-12-00443-f004:**
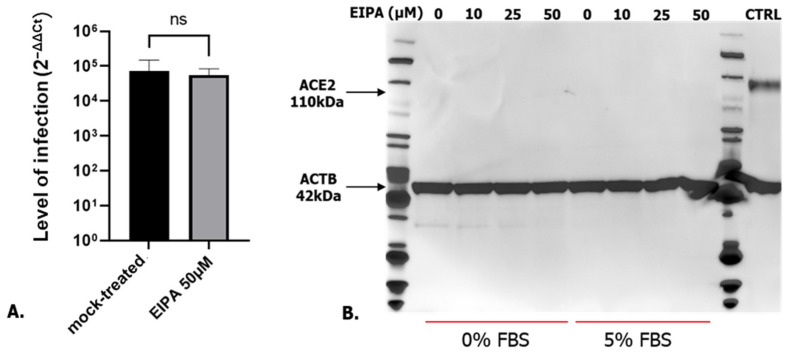
(**A**) Level of infection in mock or EIPA-treated A549 cells 1 hpi. Student *t*-test ns: non-significant. (**B**) Western blot analysis of ACE2 expression (CTRL: control, Vero E6 cell lysate).

**Figure 5 microorganisms-12-00443-f005:**
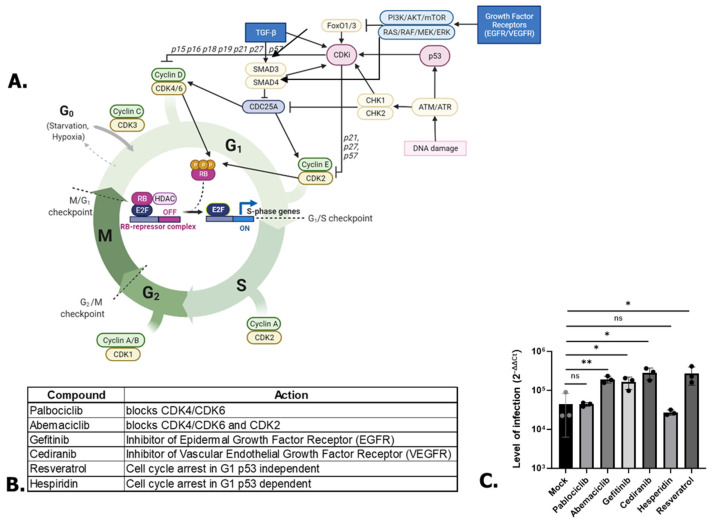
(**A**) Scheme of some proteins involved in cell cycle regulation. Created with BioRender.com. (**B**) Compounds used in the study associated with their targets. (**C**) Effect of A549 treatments on SARS-CoV-2 replication. One-way ANOVA. * *p* < 0.05, ** *p* < 0.01, ns: non-significant.

**Figure 6 microorganisms-12-00443-f006:**
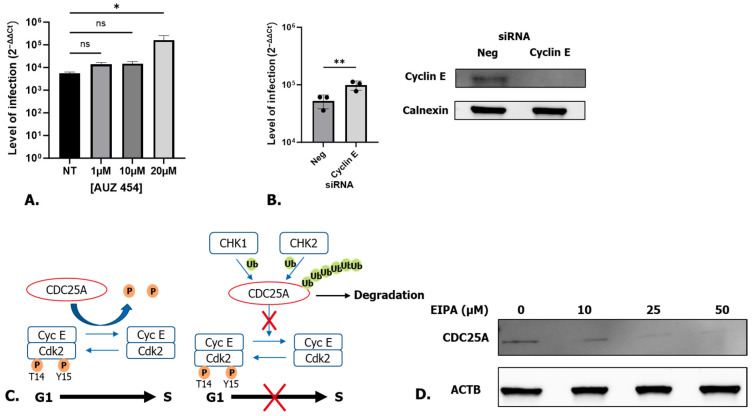
(**A**) Treatment of A549 cells with a specific inhibitor of CDK2. One-way ANOVA * *p* < 0.05 (**B**) A549 cells knocked down for cyclin E. One way ANOVA ** *p* < 0.01, ns: non-significant and Western blot analysis of cyclin E in siRNA-treated A549. (**C**) Scheme representing the role of CDC25A on cell cycle progression/arrest. (**D**) Western blot analysis of CDC25A in EIPA-treated A549 cells.

**Figure 7 microorganisms-12-00443-f007:**
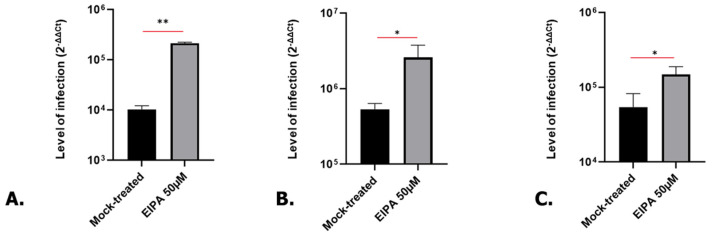
Level of infection in EIPA-treated kidney organoids ((**A**) 10^3^ PFU/organoids, 3 dpi) and ((**B**) 10^6^ PFU/organoids, 1 dpi) and lung organoids ((**C**) 10^6^ PFU/organoids, 3 dpi). Student *t*-test. * *p* < 0.05, ** *p* < 0.01.

## Data Availability

All data are available upon request.

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
