# Peer review of "P53-Independent G1-Cell Cycle Arrest Increases SARS-CoV-2 RNA Replication"

_microorganisms, 2024, doi:10.3390/microorganisms12030443_

Round 1
Reviewer 1 Report
Comments and Suggestions for Authors
This paper aims to study how the cell cycle arrest in late G1 is beneficial for SARS-CoV-2 replication showcasing that the cell cycle arrest is independent of p53 but is dependent of the CDC25A-CDK2/Cyclin E pathway.
The manuscript is written comprehensively enough to be understandable; the authors addressed this aim by demonstrating the workflow of the designed study strategy starting by talking about the four stages of the eukaryotic cell cycle and focused on the cellular pathways involved in the G0/G1 stage to showcase which pathway is important for SARS-CoV-2 replication that’s why they used several compounds blocking the cell cycle in G0/G1 in A549 cells.
The paper stated the purpose, discussion and global implication are clearly stated and consistent with the rest of the manuscript; authors provided the required tests and analysis and enough information in their discussion by using a good number of important articles talked about the subject.
The authors addressed their hypothesis and opinion in a reproducible way and they proved their results through all the required experiments and analysis, they used enough number of analyses to prove their results. The results were presented in a clear way which facilitate in reaching a conclusion elucidates that all the collected data in this study align with the literature previous data and prove the importance of the CDC25A/CDK2/Cyclin E pathway-dependent G1 cell cycle arrest in SARS-CoV-2 infection.
The authors should clarify what positive/ negative control they used in their test?
The abbreviations should be explained at the first place they are mentioned.
In vitro, in vivo, et al.: should be written in italic.
No plagiarism has been detected.
References: The authors followed the journal guidelines for some references.
Author Response
This paper aims to study how the cell cycle arrest in late G1 is beneficial for SARS-CoV-2 replication showcasing that the cell cycle arrest is independent of p53 but is dependent of the CDC25A-CDK2/Cyclin E pathway.
The manuscript is written comprehensively enough to be understandable; the authors addressed this aim by demonstrating the workflow of the designed study strategy starting by talking about the four stages of the eukaryotic cell cycle and focused on the cellular pathways involved in the G0/G1 stage to showcase which pathway is important for SARS-CoV-2 replication that’s why they used several compounds blocking the cell cycle in G0/G1 in A549 cells.
The paper stated the purpose, discussion and global implication are clearly stated and consistent with the rest of the manuscript; authors provided the required tests and analysis and enough information in their discussion by using a good number of important articles talked about the subject.
The authors addressed their hypothesis and opinion in a reproducible way and they proved their results through all the required experiments and analysis, they used enough number of analyses to prove their results. The results were presented in a clear way which facilitate in reaching a conclusion elucidates that all the collected data in this study align with the literature previous data and prove the importance of the CDC25A/CDK2/Cyclin E pathway-dependent G1 cell cycle arrest in SARS-CoV-2 infection.
The authors should clarify what positive/ negative control they used in their test?
Thanks for the comments. Mock treated (DMSO 0.4%) is used as a negative control and EIPA treated cells as a positive for all the compounds experiments. It is now better indicated in the text (lines 114-115, 173-174, 236 and 289). We also added the control of ACE2 protein expression in the western-blot figure 4.
The abbreviations should be explained at the first place they are mentioned.
Thank you for your comment. Abbreviations are now properly explained
In vitro, in vivo, et al.: should be written in italic.
Thank you. It is now corrected.
Reviewer 2 Report
Comments and Suggestions for Authors
The present manuscript data directs the attention of the scientific community to the development of antiviral agents inhibiting the replication of SARS-CoV-2 outside the p53 pathway. The authors present an effect of the CDC25A/CDK2/Cyclin E pathway as a promising approach in the SARS-CoV-2 cell cycle modulation.
The presented research includes a wide range of methods. The introduction is extensive and well presented, and the results are directly related to the discussion and represent a significant contribution to the field of COVID-19.
I have some minor recommendations:
1. It needs to be added a short "Conclusion".
2. A considerable number of grammatical errors have been discovered which to be corrected before acceptance.
Comments on the Quality of English LanguageCorrect grammatical errors.
Author Response
The present manuscript data directs the attention of the scientific community to the development of antiviral agents inhibiting the replication of SARS-CoV-2 outside the p53 pathway. The authors present an effect of the CDC25A/CDK2/Cyclin E pathway as a promising approach in the SARS-CoV-2 cell cycle modulation.
The presented research includes a wide range of methods. The introduction is extensive and well presented, and the results are directly related to the discussion and represent a significant contribution to the field of COVID-19.
I have some minor recommendations:
- It needs to be added a short "Conclusion".
Thank you for your comment. A conclusion is now added to the MS.
- A considerable number of grammatical errors have been discovered which to be corrected before acceptance.
Thank you for your comment. The manuscript is now proofread by a native English speaker and grammatical errors are now corrected.
Reviewer 3 Report
Comments and Suggestions for Authors
Fascinating work, I have some doubts.
At what point in the replicative cycle does SARS-CoV-2 arrest the cell cycle? I suggest making an infection curve.
Did they not check other regulatory proteins, such as p27 or p21?
Author Response
Fascinating work, I have some doubts.
At what point in the replicative cycle does SARS-CoV-2 arrest the cell cycle? I suggest making an infection curve.
Thank you for your comment. We agree that it would be very interesting to discover the mechanisms underlying cell cycle arrest by SARS-CoV-2 and can be the subject of a future study. The purpose of our study was to take the question the over way: how does the cell cycle arrest affect SARS-CoV-2 replication?
Did they not check other regulatory proteins, such as p27 or p21?
Thank you for this comment. We checked for p21. p21 was increased in the EIPA treated A549 compared to non-treated but not in a concentration dependent way. We think the effect observed in SARS-CoV-2 replication may involve p27 or p57. It will be tested in the future.